# Sedentary Behavioral Studies of Young and Middle-Aged Adults with Hypertension in the Framework of Behavioral Epidemiology: A Scoping Review

**DOI:** 10.3390/ijerph192416796

**Published:** 2022-12-14

**Authors:** Kexin Zhang, Sufang Huang, Danni Feng, Xiaorong Lang, Quan Wang, Yuchen Liu

**Affiliations:** 1Tongji Hospital, Tongji Medical College, Huazhong University of Science and Technology, Wuhan 430030, China; 2School of Nursing, Tongji Medical College, Huazhong University of Science and Technology, Wuhan 430030, China

**Keywords:** hypertension, sedentary behaviors, scoping review, behavioral epidemiology

## Abstract

(1) Background: As times change, the detection rate of hypertension is increasing in the young and middle-aged population due to prevalent sedentary behaviors. The purpose of this study was to conduct a scoping review to identify and summarize the research on sedentary behavior in this population by separating it into five stages: the relationship between sedentary behavior and health; measurement modalities; influencing factors; interventions; and translational research in young and middle-aged adults with hypertension. (2) Methods: Using a scoping review research approach, the PubMed, Web of Science Core Collection, and MEDLINE databases were used to search for the literature on this subject from the date of the database’s creation to 14 June 2022, and the behavioral epidemiology framework was used to classify the retrieved articles. (3) Results: A total of eight articles were included. Among them, there were six articles on the relationship between behavior and health, which includes blood pressure, insulin resistance, and the cardiovascular system; one article on the study of measurement methods, which was used for clinical decision making through decision trees; one article on influencing factors, which was divided into intrinsic and extrinsic factors; and no articles on intervention program development or the translation of intervention programs to further practice in this population. (4) Conclusions: Sedentary behavioral studies of young and middle-aged adults with hypertension are scarce and are generally carried out in the early stages of the condition. In the future, in-depth studies can be conducted on the dose–response relationship between sedentary behavior and health in this population; the development of easier and targeted measurement tools; the exploration of more influencing factors; and the effectiveness and translation of intervention programs.

## 1. Introduction

Hypertension has become an important public health problem that threatens life and health, and studies show that the age of diagnosis of hypertension is decreasing, with the number of young and middle-aged adults with hypertension already exceeding 20.0% and growing [1]. Hypertension can cause a variety of injuries to target organs, including left ventricular (LV) hypertrophy, atherosclerosis, and carotid intima–media thickness, and can further lead to type 2 diabetes mellitus, chronic kidney disease, and even other chronic cardiovascular and cerebrovascular diseases [2]. In addition, high blood pressure also affects health-related quality of life (HRQoL) [3]. At the same time, young and middle-aged people are facing huge work and family pressures, changing lifestyles, and prevalent sedentary behaviors, which increase their risk for developing multiple health issues [4]. A large number of adults (54.2%) do not meet the WHO’s recommended level of physical activity [5], especially office workers, who maintain sedentary behaviors during the workday for 82.0% of the time [6].

Surveys have shown that 31.3–74.3% of people with hypertension have sedentary behaviors or sedentary lifestyles [7,8,9,10,11,12,13,14] and that sedentary behaviors increase the level of blood pressure in adults with hypertension, increase the risk of CVD in adults [15], and negatively affect the cardiovascular system and metabolism [16,17]. Antihypertensive medication has been the mainstream therapy in recent decades, but the rate of hypertension control has not changed significantly and may be related to poor lifestyle practices [18,19], such as sedentary behaviors. The guidelines state that healthy behavior changes play an important role in hypertension prevention and blood pressure control [20] and that the use of light-to-moderate physical activity as a substitute for sedentary behavior is of great importance in promoting health in young and middle-aged adults with hypertension [21]. Sedentary behavior (SB) includes any sitting or reclining posture behavior with an energy expenditure ≤1.5 metabolic equivalents (MET) in the awake condition [22]. Previous studies have often confused sedentary behavior with physical inactivity (PI), which is defined as not achieving the guideline-recommended level of physical activity [23], but still having an activity level that is often >1.5 MET. When SB and PI coexist, it is considered a sedentary lifestyle. There is no uniform definition of a sedentary lifestyle (SL). Martins et al. [24] considered an SL to be the same concept as PI and a lack of physical activity, whereas de Leon et al. [25] suggested using less than 25–30 min of physical activity per day as the definition of SL in clinical practice. In all of the above definitions, it is considered that SL, SB, and PI exist simultaneously. Therefore, this paper continues the above concepts and defines an SL as prolonged SB and the presence of PI. In conclusion, people with sedentary lifestyles must have sedentary behaviors, and thus the concept of sedentary behavior will be used consistently in later analyses.

The American scholars Sallis et al. [26] proposed a behavioral epidemiology framework in 2000, and this framework has been widely used to analyze the research progress and development of health behaviors [27,28]. A scoping review can reveal the current state of research within a particular topic and can analyze and discuss the research in a specific area more comprehensively, helping researchers to establish a clearer, more in-depth, and more explicit perception of the current state of the field [29]. However, at present, SB-related studies are still focused on the general population, and there are relatively few reviews of research progress related to SB in the chronic disease population, especially in young and middle-aged adults with hypertension, and the current status of the research field is unclear. With the increasing number of young and middle-aged adults with hypertension, clarifying what health outcomes SB causes in this population, what targeted measurement modalities and influencing factors are available, and the availability of intervention programs should also attract the attention of researchers, as all still require more extensive analyses. Therefore, this study used this framework to initiate a scoping review of the current state of research on SB in young and middle-aged adults with hypertension to provide suggestions for subsequent research directions.

## 2. Materials and Methods

### 2.1. Research Question

Based on the behavioral epidemiological framework, SB-related studies were divided into five stages:

Stage 1 includes studies that explore and verify the association between SB and health, with the aim of investigating how SB affects health outcomes in young and middle-aged adults with hypertension. Stage 2 entails research on methods for diagnosing and quantifying SB in this population. Stage 3 includes studies addressing relevant factors affecting SB in this population. Stage 4 includes studies on intervention strategies for SB that are aimed at reducing the duration of SB or changing the type of SB in this population, and the results should be focused on behavior or biomarkers of behavioral variables. Stage 5 includes studies on the practical application of intervention strategies and on the promotion and large-scale application of intervention programs in this population.

The criteria for determining the classification were as follows: (1) The entire article was read to determine the study’s purpose and the outcome variables under investigation. (2) If the outcome variable of the study was health outcomes, the purpose of the study was to explore the effect of behavior on health outcomes, and therefore it was classified as stage one. (3) If the paper involved the measurement of behavior, it was classified as stage two; (4) If the paper involved the potential effect on behavior, it was classified as stage three. (5) Papers that addressed the effectiveness or evaluated the effectiveness of interventions on behavior were classified as stage four. (6) Papers that assessed the extent to which relevant policies or interventions were implemented and maintained were classified as stage five. (7) Papers that fit into more than one category were coded as the highest stage.

The research methodology and reporting requirements of a scoping review were used to conduct the study [29,30,31], where the reporting specifications refer to PRISMA-SCR guideline requirements [32]. The main research question of this study was, based on this framework, what is the progress in SB research in young and middle-aged adults with hypertension?

### 2.2. Inclusion and Exclusion Criteria

Inclusion criteria: (1) the literature type was original studies (qualitative studies, quantitative studies, mixed studies, etc.); (2) the article was written in English; and (3) the study topic was SB in young and middle-aged adults with hypertension, with the age range of the study subjects being 18–65 years.

Exclusion criteria: (1) studies on children, the elderly, hypertensive pregnancies, or people without hypertension; (2) non-original studies (reviews, systematic reviews, scoping reviews, meta-analyses, commentaries, book reviews, etc.); (3) conferences, books, unpublished articles, and dissertations; (4) studies on SB topics that do not focus on young and middle-aged people with hypertension (e.g., the primarily study physical activity, the research focus is not SB, etc.); (5) full text not available; and (6) articles not applicable to analyses within this behavioral epidemiological framework (e.g., articles reporting only on the current status of SB in young and middle-aged people with hypertension).

### 2.3. Literature Sources and Search Strategies

English articles were obtained by using the PubMed, Web of Science Core Collection, and MEDLINE databases to search the literature from the date of the database’s creation to 14 June 2022. The search was performed as an advanced search using MeSH terms and keywords. According to the MeSH word list, SL was included in this study as an SB subtopic term in the review.

The English search terms were: (1) (Adults) OR (Young Adult) OR (Middle Aged); (2) (Sedentary behavior) OR (Behavior, Sedentary) OR (Sedentary Behaviors) OR (Sedentary Lifestyle) OR (Lifestyle, Sedentary) OR (Physical Inactivity) OR (Inactivity, Physical) OR (Lack of Physical Activity) OR (Sedentary Time) OR (Sedentary Times) OR (Time, Sedentary); and (3) (Hypertension) OR (Blood Pressure, High) OR (Blood Pressures, High) OR (High Blood Pressure) OR (High Blood Pressures). The groups were connected in parallel using “AND”.

### 2.4. Evidence Selection and Information Extraction

The resulting literature captions were extracted and imported into NoteExpress, and duplicate articles were excluded. Two researchers (K.Z. and D.F.) read the article titles, abstracts, and keywords according to the inclusion and exclusion criteria and performed the initial screening independently. After the initial screening was completed, the remaining full-text articles were read and screened again to determine the final included literature. The full texts of the included literature were read through and classified according to the main outcome indicators and the research foci of the articles in relation to the behavioral epidemiological framework stages. Throughout the process, both researchers were independently and professionally trained, and a third researcher (Q.W.) was consulted in the case of a disagreement. The final article authors (time of publication), sample, country/region, blood pressure level, study type, measurement tools, and outcome indicators were extracted. The articles were categorized according to the behavioral epidemiological framework of the stage they were in and were analyzed in stages, as shown in Table 1.

## 3. Results

### 3.1. Literature Search Results

A total of 6056 articles were retrieved from the database according to the search strategy, and 2 articles were added by reading other articles’ references [8,43], bringing the total to 6058 articles. There were 1951 duplicate articles removed; 3918 documents unrelated to the topic were excluded; and 18 articles from conferences, books, dissertations, etc., were excluded, leaving 171 articles. After reading the full text of the remaining articles and rescreening, a total of 163 documents were excluded (non-original studies (reviews, book reviews, etc.) and non-English, *n* = 13; study population not young and middle-aged adults with hypertension, *n* = 35; unrelated to topic, *n* = 106; and study only reported current status of sedentary behavior, *n* = 9), and 8 documents were included [33,34,35,36,37,39,40,41]. The steps of the literature search are shown in Figure 1.

### 3.2. Characteristics of the Included Literature

According to this framework [26], the distribution of SB study topics in young and middle-aged adults with hypertension is shown in Figure 2. After searching, a total of eight SB studies on young and middle-aged adults with hypertension were included, with the highest number of studies concentrated in stage 1. Out of all the included studies, 37.5% were from the United States, with three articles written by Gibbs [33,34,35]. Six studies focused on subjects with prehypertension, hypertension stage 1, and hypertension stage 2. There was no literature included that met stage 4 (studies on intervention strategies for SB) or stage 5 (studies on the practical application of intervention strategies). The included literature, all of which were quantitative studies, included prospective studies (25.0%), cross-sectional studies (37.5%), and randomized controlled trials (37.5%).

### 3.3. Inclusion of Details from the Literature

#### 3.3.1. Relationships with Health

A total of six (75.0%) studies were classified as stage 1, as they analyzed the association between SB and health outcomes in young and middle-aged adults with hypertension (one cross-sectional study, two prospective studies, and three randomized controlled trials), as shown in Table 1. The reason for classifying these randomized controlled trials [33,34,35] as stage 1 rather than stage 4 is that, by reading the full articles, it was clear that the primary foci of the studies and the purposes of the interventions in these articles were to explore the health-related changes caused by changing SB, but the foci of the studies were not on how to change SB. Changing SB was only used as a precondition in these articles. The fourth stage focuses on more general interventions for SB, and the focus of a study should be on whether the intervention program changes SB rather than health outcomes.

After controlling for demographic variables, it was found that people with hypertension who displayed SB had lower insulin sensitivity and greater insulin resistance [36]. A randomized controlled trial showed that interrupting prolonged SB with intermittent standing significantly reduced diastolic blood pressure (DBP) and mean arterial pressure (MAP) in subjects compared to continuous SB [33]. Alansare et al. [34] found that maintaining SB after a simulated workday experiment resulted in increases in sitting systolic blood pressure (SBP), DBP, MAP, supine carotid–femoral pulse wave velocity (cfPWV), and carotid–ankle pulse wave velocity (caPWV). In people with elevated blood pressure, prolonged SB maintenance causes a significant slowing of the cerebral blood flow velocity (CBFv) in the middle of the day [35]. Prospective studies have shown a higher risk of new-onset LV hypertrophy in adults with hypertension and an SL compared to physically active people, with a significant increase in the LV end-diastolic diameter [39]. In a study by Palatini et al. [37], they showed that the cumulative maximum and cumulative mean carotid intima–media thickness (IMT) of the three carotid segments was higher in subjects with an SL than in subjects with adequate activity.

#### 3.3.2. Measurement Method

In the second stage, one study (12.5%) focused on the clinical diagnosis and prediction of SB in young and middle-aged adults with hypertension, as shown in Table 1.

Castelo et al. [40] used the CHAID algorithm with a decision tree model to construct a decision tree for SB in adults with hypertension to analyze the probability of SB in individuals with a sedentary lifestyle. The first node of this decision tree model was the clinical indicator “daily physical inactivity”, and the probability of SB was 0.88 for those with this indicator, whereas the second node was “physical inactivity in leisure time” and the probability of SB was 0.99 for those with both of these two indicators. This decision tree has a predictive power of 69.5% and can be used by nurses in the clinic to diagnose SB in adults with hypertension and optimize the diagnosis time.

#### 3.3.3. Influencing Factors

One (12.5%) study was included in the third stage on factors influencing SB, as it focused on respondents’ intrinsic and extrinsic factors, as shown in Table 1.

Martins et al. [41] indicated that respondents’ intrinsic factors included a “lack of motivation to perform physical activity”, a “lack of interest in physical activity”, “mindset, beliefs, and health habits that hinder physical activity”, and a “lack of confidence in physical activity practice”, whereas their external factors were a “lack of training to complete physical activity” and a “lack of physical activity training”.

The study also pointed out that the factors of a “Lack of motivation to perform physical activity” and a “lack of interest in physical activity” were the most important reasons for the existence of SB (PR = 5.358), and the presence of these factors increased the probability of an SB diagnosis five-fold [41].

## 4. Discussion

### 4.1. SB and Health

SB is detrimental to the short- and long-term health of young and middle-aged adults with hypertension.

Based on the stage 1 study, it is evident that SB negatively affects blood pressure (blood pressure level and blood pressure management), the cardiovascular system (left ventricular hypertrophy, carotid intima–medial thickness, cerebral blood flow velocity, and pulse wave velocity), and insulin sensitivity in young and middle-aged adults with hypertension. Prolonged SB increases the degree of blood pressure in young and middle-aged people with hypertension and is detrimental to their blood pressure control [33]. In the long term, its effects on the carotid intima–medial thickness and pulse wave velocity in this population reflect its adverse effects on vascular health [37], further increasing the risk of CVD, such as atherosclerosis, and a poor prognosis [15]. This was also reflected by outcomes such as left ventricular hypertrophy in the prospective study [39]. Additionally, the effect of SB on insulin sensitivity in this population cannot be ignored, which may indicate that prolonged exposure to SB increases the risk of diabetes in this population [36].

Prolonged SB during the workday increases the degree of sitting blood pressure in adults with hypertension [34]. In contrast, SB was also significantly associated with the development of hypertension in a non-hypertensive young and middle-aged population [44,45]. Koyama et al. [46] and Vancampfort et al. [47] noted that reducing sedentary time in patients with chronic diseases was more beneficial in improving their prognosis and quality of life than exercising alone. Other studies have noted that SB has the strongest association out of all factors associated with all causes of death from CVD [48,49].

Therefore, there is a need to accurately measure SB in young and middle-aged adults with hypertension to identify influencing factors and explore effective interventions to improve prognoses. Although the stage 1 inclusion study pointed out that SB may contribute to cardiovascular disease risk and adverse long-term health outcomes in young and middle-aged adults with hypertension, the dose–response relationship between SB and adverse health outcomes in this population is not clear. In addition, studies have focused on overall SB duration and the different health effects that can result from different SBs remain understudied.

### 4.2. Diagnosis and Measurement of SB

Diagnostic and measurement tools for SB are poorly researched and of limited utility. The accurate diagnosis and measurement of SB is a prerequisite for the identification of SB and its extent.

Although decision trees have shown a high accuracy in the identification and diagnosis of SB, the algorithm used is deficient in terms of stability and is susceptible to overfitting problems due to factors such as the number of nodes and is not suitable for extrapolation to other samples or populations [50].

Current measurements for SB are similar to those for physical activity and are divided into subjective and objective categories. Subjective measures include measurements using self-reported tools such as questionnaires and SB logs, and objective measures includes the use of tools such as accelerometer heart rate monitors [51].

Although some studies with targeted instruments have been conducted in people with hypertension, the measurement of SB in this population is still commonly reported using self-reported questionnaires [52,53]. Self-reported questionnaires are the most commonly used tool to assess physical activity, and they are also used to explore the SB of populations. The advantages are that they are less resource-intensive, more conducive to surveys in less resourced areas, and easier to replicate, but these measures are more subjective in nature [54]. Additionally, there is less evidence of applicability with more targeted questionnaires, such as the Adult Sedentary Behaviour Questionnaire (ASBQ) [55], in young and middle-aged people with hypertension. Therefore, more studies using and validating these instruments should be reported.

### 4.3. Influencing Factors of SB

The influencing factors of SB have been widely explored but are not yet systematized. Identifying influencing factors is a prerequisite for the development of better interventions. Studies have pointed out that knowledge/attitude factors are important influences on SB [41,56]. Among them, perceptions about exercise and health were examined by several studies and were significantly associated with SB. Negative health perceptions and insufficient knowledge about physical activity and SB can make young and middle-aged adults with hypertension reluctant to improve their SB, and thus it is especially important to emphasize the negative effects of SB and help young and middle-aged adults with hypertension understand this when educating them [14,41]. Unlike in the general population, the source of health information for young and middle-aged adults with hypertension is more likely to be medical personnel [14]. This suggests that healthcare workers are key participants in improving the lifestyles of adults with hypertension. In addition to the influence of knowledge and attitudes, other influencing factors include the lack of external conditions related to physical activity conditions, such as time and space [41]. Gender, age, the presence of a spouse, education level, and underlying medical conditions were also found to be influential factors for SB in people with hypertension [57,58]. Unlike other age groups, SB in the young and middle-aged population is also influenced by the type of work [59], which also requires research attention.

Overall, there is still a dearth of data, and therefore more studies examining the impact of various factors on SB in young and middle-aged adults with hypertension are required (e.g., studies on demographic factors, peer support, family influence, physician influence, etc.). By examining the knowledge and attitudes about SB, the environment for physical activity, and the community support to guide the design of intervention programs, follow-up studies should increase the use of theory.

### 4.4. General Analysis

The majority of the articles included in this study focused on subjects with prehypertension, hypertension stage I, and hypertension stage II [33,34,35,36,37,39], which may be related to the risk of physical activity and high blood pressure. There were no theory-based studies on SB in young and middle-aged adults with hypertension in the included literature. Follow-up studies could use mixed studies and qualitative studies to combine more theoretical frameworks to comprehensively and thoroughly explore the research on SB improvements and interventions in young and middle-aged people with hypertension to complement the results of the quantitative studies.

### 4.5. Advantages and Limitations

Based on a behavioral epidemiological paradigm, this paper used a scoping review approach to describe the development of SB research in middle-aged and young adults with hypertension and offers workable directions for further in-depth studies.

The following are some of the paper’s limitations: (1) Only studies published in English were included. (2) In addition, most of the studies were conducted in developed countries, which may affect the generalization of the data to populations in other countries or regions. (3) Based on the research requirements of the scoping review, no quality evaluation or risk of bias assessment of the included literature were performed, and other subsequent studies can be explored in depth through research methods such as systematic reviews.

## 5. Conclusions

Based on a behavioral epidemiological framework, this article compared and summarized the current state of research on sedentary behaviors in young and middle-aged adults with hypertension by using a scoping review approach to analyze the English literature published before 14 June 2022.

Research on SB in young and middle-aged people with hypertension is still in its infancy, and the majority of studies have concentrated on the harmful effects of SB with regard to blood pressure, the cardiovascular system, and metabolism in young and middle-aged adults with hypertension. There is still a general dearth of theoretical framework supports for studies on the factors driving SB, with the majority of studies concentrating on knowledge, attitudinal, and extrinsic factors. There is also a lack of simple and accurate measurement tools for SB, and there are no evidence-based, effective interventions for the time being. Follow-up studies could select the dose–response relationships between different SBs and health outcomes in young and middle-aged adults with hypertension, develop simpler and targeted measurement tools, explore more SB-influencing factors in combination with theories, and conduct intervention program design and translation studies.

## Figures and Tables

**Figure 1 ijerph-19-16796-f001:**
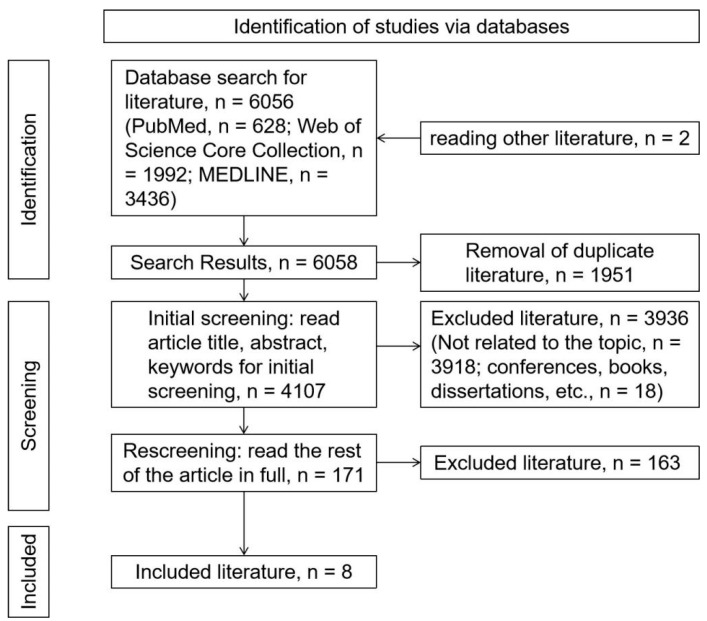
The literature search process.

**Figure 2 ijerph-19-16796-f002:**
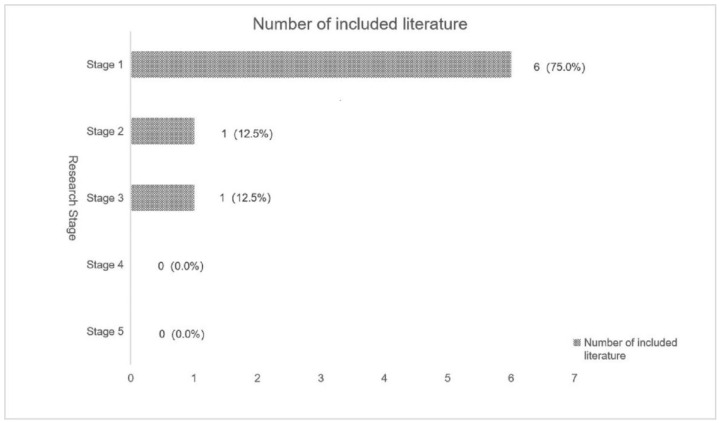
Distribution of SB study topics in young and middle-aged adults with hypertension (stage 1: studies to explore and verify the association between SB and health; stage 2: studies on SB measurement methods; stage 3: studies to explore the factors influencing SB; stage 4: studies on intervention strategies for SB; stage 5: studies on the practical application of intervention strategies).

**Table 1 ijerph-19-16796-t001:** Basic characteristics of the literature for inclusion in this analysis.

Author (Time of Publication)	Stage ^1^	Number of Samples (Age ^2^)	Country/Region	Research Type	Tools for Measuring SB (Definition/Program ^3^)	Result Indicators
Gibbs et al. [33] (2017)	1	25 (average age: 42; 20–65 years old)	United States of America	Randomized controlled trials	activPAL3 micro (two periods of 3 h and 40 min of continuous seated desk work).	BP ^4^; PWV ^5^
Alansare et al. [34] (2020)	1	25 (20–65 years old)	United States of America	Randomized controlled trials	activPAL3 micro (except for required bathroom breaks, participants worked at a conventional desk while sitting still during the sitting period, with their feet flat on the floor).	BP; PWV
Perdomo et al. [35] (2019)	1	25 (average age: 42; 20–65 years old)	United States of America	Randomized controlled trials	Global Physical Activity Questionnaire (the SB group was required to sit continuously).	BP; CBFv ^6^
Hwu et al. [36] (2004)	1	872 (35–60 years old; SB subjects averaged 51.8 ± 8.5 years old; non-SB subjects averaged 52.7 ± 8.8 years old)	Asia-Pacific	Cross-sectional studies	Total physical activity was measured by keeping track of how many hours a day were spent engaging in physical activity. (Sedentary activity hours)/(24 h—base activity hours) was used to calculate physical inactivity (participants were referred to as “SB” if their physical activity score was higher than 0.5).	Insulin resistance
Palatini et al. [37] (2011)	1	87 (SB subjects averaged 33.2 ± 8.7 years old; non-SB subjects averaged 29.7 ± 8.6 years old)	Italy	Prospective studies	A standardized questionnaire was used to measure physical activity [38] (participants were categorized as sedentary if they did not regularly engage in any physical activity).	Carotid intima–media thickness
Palatini et al. [39] (2009)	1	454 (18–45 years old; average age: 33.1 ± 8.4 years old)	Italy	Prospective studies	A standardized questionnaire was used to measure physical activity [38] (participants were categorized as sedentary if they did not regularly engage in any physical activity).	LV ^7^ mass; LV hypertrophy
Castelo et al. [40] (2016)	2	285 (19–59 years old)	Brazil	Cross-sectional studies	An instrument was used with variables related to identification, related factors, and signs and symptoms. The diagnostic inference was made by four nurses who had received prior training in an 8-h course [40] (definition based on NANDA-I ^9^).	SB ^8^
Martins et al. [41] (2015)	3	285 (19–59 years old)	Brazil	Cross-sectional studies	A tool that was created based on previously validated clinical signs and related factors was used to collect the data [42] (individuals with SB were defined as those who do not exercise at the recommended frequency, duration, and intensity or do not expend significant energy to improve their physical condition).	SB; PA ^10^

^1^ Stage: Classification of study stages based on an epidemiological framework. Stage 1: studies to explore and verify the association between SB and health; stage 2: studies on SB measurement methods; stage 3: studies to explore the factors influencing SB; stage 4: studies on intervention strategies for SB; stage 5: studies on the practical application of intervention strategies. ^2^ Age: organized according to the age ranges of the subjects included in the studies. ^3^ Definition/program: the definition of SB used in the study or the setting of SB in the study protocol. ^4^ BP: blood pressure. ^5^ PWV: pulse wave velocity. ^6^ CBFv: pulse cerebral blood flow velocity. ^7^ LV: left ventricular. ^8^ SB: sedentary behavior. ^9^ Definition based on NANDA-I: “Choose a daily routine without exercise”; “It shows a lack of physical fitness”; “Verbalizes preference for activities with little exercise”. ^10^ PA: physical activity.

## Data Availability

Not applicable.

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
