# Peer review of "Sedentary Behavioral Studies of Young and Middle-Aged Adults with Hypertension in the Framework of Behavioral Epidemiology: A Scoping Review"

_ijerph, 2022, doi:10.3390/ijerph192416796_

Round 1
Reviewer 1 Report
This is an important topic regarding sedentary behaviors in young and middle-aged adults with hypertension and this review can be significant contribution to the literature. The introduction section is well presented. I also like the use of the behavioral epidemiology framework to guide the review. The selection of scoping review is the right review method for the research question. However, a few areas can be improved:
My major comments:
Introduction section –
· This section is well written. However, it would be improved by adding the current state of SB in this population since you are not including this important information according the behavioral epidemiology framework.
Methods section –
· It’s important to cite the methodology paper for scoping review which was used to guide this scoping review
· Please also refer to PRISMA reporting guideline for reporting a review paper like this.
o Please add more detail regarding your data analysis and synthesize process. For instance, how did you decide which article belong to which stage of the behavioral epidemiology framework?
o Line 94 – “additions were made….” This sentence is not making sense
o Line 98 – adding “adults …..” as a group of search term would be potentially problematic as you can miss other articles that do not use this terms, for instance, mothers, workers, etc. No sure if you have tested your search strategy with a preliminary set of articles.
o Figure 1 is of low picture quality. Also, please refer to PRISMA flow chart for report the study selection process
o Have you registered your review protocol anywhere?
· Please add quality assessment for all selected studies
Results section –
· Line 126 – these reasons for exclusion came as a surprise as they were not mentioned in your study inclusion or exclusion criteria in methods section. Also, what do you mean by “unrelated topic”?
· Table 1 – It was not clear why these RCTs were put in stage 1 instead of stage 4 according the behavioral epidemiology framework. They all were testing effective of some interventions. I get that they were probably done in labs in a very controlled setting to compare SB vs. not SB on the health outcomes, but the decision-making process was not explicit.
· Line 220 – the narratives in this paragraph was not consistent with the paragraph below. Since you have only one study included, all the findings should come from the same study.
Discussion section –
· The first two points of limitation section should be moved to in exclusion criteria.
· The third point of limitation is something you can address by revising the manuscript. So please revise and do the quality assessment of selected articles.
· In general, in addition to discuss each stage separately, there should be more synthesize of the finding by having an overall discussion, including addressing discussing the current SB status of the population and the current state of research in SB for other similar or relevant population.
Other comments:
· Line 332 – it is a problem to me that only one person is involved in the formal data analysis stage; the data extraction process should involve at least two reviewers, and the data synthesize process should involve a team discussion at least.
Minor comments:
· Please change the wording the title: “hypertensives” is not a very patient-centered word, I would use “adults with hypertension”
· Line 226 – “on the hand”, I don’t quite get why this phrase is there…
Author Response
Dear Reviewer,
Thank you for your comments. Please see the detailed response in the attachment.

Reviewer 2 Report
Title
I suggest changing the title to: Sedentary behavioral studies of young and middle-aged hypertensives in the framework of behavioral epidemiology: A scoping review
Introduction
Considering that the purpose of the study is not to explain the conceptual difference between a sedentary lifestyle, sedentary behavior and physical inactivity, I suggest deleting the section between lines 52 to 62.
Standardize the percentage values ​​(I suggest a place after the comma).
The justification presented for carrying out the study is fragile. In addition, several objectives presented in this review have been explored previously. So, what makes this study different from others that have already been published? I suggest mentioning other aspects that may highlight the importance of this review.
Methods
Line 90 - Exclusion criteria: In item (2) non-original studies, exclude etc and inform all types of excluded studies.
Line 91 - According to item (4) studies with full text not available were excluded. Were unavailable full-text studies identified? Was there an attempt to contact the authors of these studies to try to obtain these studies? I suggest mentioning.
Lines 107 and 111 – mention, in parentheses, the initials of the researchers who participated in these stages (eg: KZ; DF)
Results
Line 119 – The authors inform that 2 studies were included after reading other literature. Inform what this literature was.
Line 120 – Of the studies identified, were 1951 duplicates? I suggest checking this information.
Line 152 – I believe that part of this excerpt could be included in the discussion, as it addresses a possible explanation of part of the findings.
Table 1: I suggest replacing the title of the first column with the term Author.
Tables 2, 3 and 4 each cite only one study. So I suggest deleting these tables and entering the information in text format.
Line 172 – Authors mention studies, but cite only one reference. Adjust.
Discussion
Line 233 – The elements presented in item 4.1. are fragile and do not deepen the relationship of the review findings with the literature.
Line 255 – Have the tools for diagnosis and measurement of BS been little explored in the literature or was this conclusion established solely based on the findings of this review?
I suggest including in the topic 4.5 Advantages and Limitations
Only studies published in English were included. In addition, all studies were carried out in developed countries, which could influence the generalization of data to other population groups.
Author Response

(The authors gave the same response as above.)

Reviewer 3 Report
Thank you for the opportunity to review this manuscript conducting a scoping review of sedentary behavior research in this population in five stages: the relationship between sedentary behavior and health; measurement modalities; influencing factors; interventions; and translational research in young and middle-aged hypertensives to identify and summarize research progress. This reviewer has certain questions, which should be addressed by authors prior to publications.
Was this review registered on PROSPERO?
Was the risk of bias assessment performed?
The inclusion and exclusion criteria were unclear. On line 122-127, study including multimorbidity coexistence, study topic was physical activity, and study only reported current status of sedentary behavior were excluded after full text review. However, they are inconsistent with exclusion criteria in the methods section. Furthermore, 4 cross-sectional studies were included, despite study only reported current status of sedentary behavior were excluded.
It is unclear whether all subjects included in this review were 18-65 years old or not (for example, reference 32).
The definition of sedentary behavior in this review is not clear. On line 172 and 195 in the results section, sedentary lifestyle (SL) was used. It should be clearly defined in the methods section (inclusion criteria).
It would be better to show the definition of sedentary behavior, and how to assess sedentary behavior in each study in the Tables 1-4.
On line 270-272, please insert the references.
Author Response

(The authors gave the same response as above.)

Reviewer 4 Report
The main purpose of this review was to investigate the effect of sedentary behavior on the development of arterial hypertension in young and middle-aged people.
This research topic is relevant in the field of environmental pathophysiology, behavioral epidemiology. The data obtained help to close the gap in the field of the etiology of arterial hypertension in young and middle-aged people.
Since studies of sedentary lifestyle among young and middle-aged hypertensive patients are rare, the review outlines the prospects for studying the dose-response relationship between a sedentary lifestyle and health among this age group of the population. The problem of finding available simple and effective measuring tools to assess the impact of a sedentary lifestyle on the risk of arterial hypertension is also important.
The conclusion is consistent with the evidence and arguments presented and answers the main question posed.
Literature references are generally appropriate for the tasks at hand
The text may need some literary editing
Author Response
Dear Reviewer
Thank you for your comments and suggestions. We have made some improvements in the revision process to address these issues.
Round 2
Reviewer 1 Report
Thanks for addressing my previous comments. I just have two minor comments.
Line 44 - change “hypertensive people” to “people with hypertension”. Please note that in diabetes related clinical practice and research, the word “diabetic” is deemed as un-respectful to the patient; similarly, in your manuscript, please avoid using the language like “hypertensive people”, and change it to “people with hypertension” throughout the manuscript
Line 87 – 95 – I like that you added more explanation in each stage. It would be helpful to also add the potential study designs for each stage. I like your response to my “point 10” in my earlier comments. Please note that for intervention studies targeted on behavior change, many of the studies may also include health or biomedical or biobehavioral related outcomes, in addition to behavioral change related outcomes.
Author Response

(The authors gave the same response as above.)

Reviewer 2 Report
Point 3: Standardize the percentage values. Adjust line 35 (20.0%) and line 43 (82.0%).
Author Response
Dear Reviewer,
Thank you for your suggestion. We have standardized the percentages.
Reviewer 3 Report
The authors have addressed the issues that I raised, and the manuscript is significantly better than before. However, it is still unclear how to assess and define sedentary behavior in each included study (e.g. the name of self-reported questionnaires, and the cut-off point). More detailed information would be helpful for readers to make sense of this scoping review.
Point 6: It would be better to show the definition of sedentary behavior, and how to assess sedentary behavior in each study in the Tables 1-4.
Response 6: Thank you for your comments, the section has been revised in the text.
Author Response
Dear Reviewers
Thank you for your suggestion. In response to your suggestion, we have clarified the column "Tools for Measuring SB (Definition/Program 3)" in Table 1 and added the corresponding content.